Optimized customer churn prediction using tabular generative adversarial network (GAN)-based hybrid sampling method and cost-sensitive learning

http://orcid.org/0009-0005-5458-5125 Adiputra I Nyoman Mahayasa 1
Wanchai Paweena 1 wpaweena@kku.ac.th
http://orcid.org/0000-0003-0735-2693 Lin Pei-Chun 2
1 College of Computing, Khon Kaen University , Khon Kaen , Thailand
2 Department of Information Engineering and Computer Science, Feng Chia University , Taichung , Taiwan
Alatas Bilal
Electronic publication date: 2025 Jun 19
Publication date: 2025
Volume: 11
Electronic Location ID: e2949
Received 2024 Nov 14; Accepted 2025 May 19
Copyright: © 2025 Adiputra et al.
Copyright year: 2025
Copyright holder: Adiputra et al.
License: This is an open access article distributed under the terms of the Creative Commons Attribution License, which permits unrestricted use, distribution, reproduction and adaptation in any medium and for any purpose provided that it is properly attributed. For attribution, the original author(s), title, publication source (PeerJ Computer Science) and either DOI or URL of the article must be cited.
License URL: https://creativecommons.org/licenses/by/4.0/

Keywords: Customer churn prediction, GAN-based hybrid sampling method, Cost-sensitive learning

Funding: Khon Kaen University This study was supported by Khon Kaen University on ASEAN GMS grant. The funders had no role in study design, data collection and analysis, decision to publish, or preparation of the manuscript.

==============================
Background

Imbalanced and overlapped data in customer churn prediction significantly impact classification results. Various sampling and hybrid sampling methods have demonstrated effectiveness in addressing these issues. However, these methods have not performed well with classical machine learning algorithms.

Methods

To optimize the performance of classical machine learning on customer churn prediction tasks, this study introduces an extension framework called CostLearnGAN, a tabular generative adversarial network (GAN)-hybrid sampling method, and cost-sensitive Learning. Utilizing a cost-sensitive learning perspective, this research aims to enhance the performance of several classical machine learning algorithms in customer churn prediction tasks. Based on the experimental results classical machine learning algorithms exhibit shorter execution times, making them suitable for predicting churn in large customer bases.

Results

This study conducted an experiment with six comparative sampling methods, six datasets, and three machine learning algorithms. The results show that CostLearnGAN achieved a satisfying result across all evaluation metrics with a 1.44 average mean rank score. Additionally, this study provided a robustness measurement for algorithms, demonstrating that CostLearnGAN outperforms other sampling methods in improving the performance of classical machine learning models with a 5.68 robustness value on average.

Introduction

Customer churn is the state that customers no longer use the services of the company. Predicting customer churn is essential for businesses to reduce costs and drive growth (Faraji Googerdchi, Asadi & Jafari, 2024). The main objective of churn analysis is to detect and predict potential customer churn at an early stage, enabling companies to address their concerns effectively. This will aid in fulfilling customer needs, ensuring customers remain satisfied and continue using the service (Wagh et al., 2024). The main problem of customer churn prediction is imbalanced and overlapped data (Googerdchi, Asadi & Jafari, 2024; Verhelst et al., 2023). Imbalanced data occurs when the number of instances in the classes is significantly different from the other classes. In customer churn predictions the numbers of churn class or the customers who leave company service are usually not as much as customers who stayed on company service. Overlapped data occurs when the data of one class crosses to the other region of the class after applying an oversampling technique (Vuttipittayamongkol & Elyan, 2020; Zhu et al., 2022). Oversampling is not the only reason that caused overlapped data. Classes may naturally overlap due to the similarity in feature distributions. Instances from different classes appear similar, leading to overlapping regions in the feature space. As a result of that condition, machine learning algorithms failed to deliver satisfactory performance in predicting customer churn.

In recent years, there have been several research studies aimed at overcoming the problem of imbalanced and overlapped data in customer churn prediction. The method is called a hybrid data-level solution (Ding et al., 2023; Lu, Cheung & Tang, 2016; Ouf, Mahmoud & Abdel-Fattah, 2024; Xu et al., 2020; Zhu et al., 2022). The method combines oversampling techniques such as Synthetic Minority Oversampling Technique (SMOTE) and generative adversarial network (GAN) and under-sampling techniques such as edited nearest neighbor (ENN) and Tomek-links.

Several research works on cost-sensitive learning (CSL) have been done to overcome the data imbalanced problems in machine learning classification tasks (Kumaravel & Vijayan, 2023; Mienye & Sun, 2021; Vanderschueren et al., 2022; Zhou et al., 2023). Unlike traditional classification, CSL assigns varying costs to different types of errors. It enables learning algorithms to adapt to imbalanced data distributions without changing the underlying learning principles. CSL addresses imbalanced issues by incorporating different penalties or weights for misclassifying instances from different classes, thereby encouraging the model to pay more attention to the minority class.

Customer churn prediction is a critical task for businesses aiming to retain customers and sustain long-term profitability (Wen et al., 2022). However, the inherent class imbalance in churn datasets poses a significant challenge, as traditional machine learning algorithms often struggle to learn from the minority class, leading to suboptimal predictions.

The challenge of imbalanced data in customer churn prediction lies in generating realistic synthetic samples that accurately reflect the minority class distribution. Traditional oversampling methods, such as SMOTE and its variants, rely on linear interpolation, failing to capture the complex, nonlinear patterns inherent in real-world data (Zhu et al., 2022). Consequently, they often produce redundant or unrealistic samples, leading to overfitting and poor generalization.

While cost-sensitive learning mitigates class imbalance by penalizing misclassifications, it does not modify the data distribution or improve the representation of the minority class, limiting its effectiveness in diverse classification scenarios (Kumaravel & Vijayan, 2023). Previous studies on GAN-based oversampling have primarily focused on data augmentation without integrating cost-sensitive learning, leaving a critical gap at the algorithm level. As a result, class imbalance persists, affecting generalization across different classifiers and constraining the full potential of GAN-generated data (Ding et al., 2023).

To address these limitations, CostLearnGAN is introduced as a hybrid framework that integrates GAN-based data augmentation with cost-sensitive learning at both the data and algorithm levels. Unlike conventional methods, CostLearnGAN leverages GANs to learn the complex data distribution, generating high-quality, diverse synthetic samples that more accurately reflect real-world minority class patterns. Additionally, by incorporating cost-sensitive learning into the classifier training process, the framework ensures a balanced learning process, reducing bias and improving model robustness across different datasets and classifiers. By bridging data-level and algorithm-level solutions, CostLearnGAN represents a significant step forward in overcoming the limitations of previous GAN-based approaches for customer churn prediction.

Through extensive experiments, this study demonstrates how CostLearnGAN not only addresses data imbalanced and overlapped problems but also fine-tunes model parameters to achieve superior generalization in customer churn prediction. By addressing the challenge of imbalanced data from both a data-generation and algorithmic perspective, CostLearnGAN represents a significant advancement over existing oversampling and cost-sensitive approaches.

The rest of this article is structured as follows. “Related Works” provides a systematic review of related work on the data-level, algorithm-level, and hybrid approach of classification tasks. In “Materials and Methods”, the baseline methods of the CostLearnGAN framework are described in detail. “Experiments” presents the experimental results along with relevant discussions. Finally, “Results” summarizes the contributions of this study.

Related works

Data-level solution

An oversampling method works by adding a new synthetic sample on minority data to boost the learning of the classifier. SMOTE is a popular method for addressing class imbalance by generating synthetic samples for the minority class. It works by selecting an example of a minority class, finding its k-nearest neighbors, and interpolating between them to create new synthetic samples. Because of its limitations, SMOTE and ENN hybrid methods were introduced. ENN removes noisy or borderline instances from both the majority and minority classes by eliminating samples that differ from most of their k-nearest neighbors. This combined approach enhances class balance while also improving data quality (Xu et al., 2020). Another SMOTE-based approach, data distribution and spectral clustering-based SMOTE (DDSC-SMOTE) (Li & Liu, 2024), operates using three strategies: adaptive allocation of synthetic sample quantities, seed sample adaptive selection, and synthetic sample improvement SMOTE has a drawback in that it blindly synthesizes new samples without considering the distribution of the imbalanced data, potentially leading to the generation of incorrect and unnecessary instances. Based on the SMOTE limitation method called the critical pattern supported three-way sampling method (CPS-3WS), this method works by evaluating the risky majority patterns to be eliminated and selecting critical minority patterns to synthesize new samples under local information constraints (Yan et al., 2024). Their experiments surpass several SMOTE-based sampling performances on average. SMOTE and another neighborhood-based limitation are not generating synthetic samples by learning the true distribution of the data, so the generated data probably does not represent the real data. Recently, the generative artificial intelligence (AI) methods that are based on neural networks are useful on synthetic data generators as well, several studies were conducted by GAN-based method. Classifier-aided minority augmentation generative adversarial network (CMAGAN), a GAN method with an outlier elimination strategy was applied to each class to minimize the negative impacts of outliers and utilized Mahalanobis distance to ensure they fall within the desired distribution (Wang, Liu & Zhu, 2024). CMAGAN generally demonstrates superior performance and delivers higher-quality augmentation results based on their experiments. A GAN-based hybrid method in extension with an under-sampling technique called GAN-based hybrid sampling (GBHS) to handle overlapped data produced by GAN (Zhu et al., 2022). GBHS significantly reduced the influence of class overlap and outperformed several customer classification dataset benchmarks. Oversampling the multi-features on tabular data like customer churn data is challenging, one of the possible solutions is Conditional Tabular GAN (CTGAN), which uses a conditional generator that allows it to handle these challenges by conditioning the generation process on specific columns, ensuring that both the categorical and continuous data distributions are well-represented (Xu et al., 2019). A hybrid sampling method CTGAN-ENN was proved to enhance the quality of synthetic data by reducing overlapping data and balancing the customer churn data class (Adiputra & Wanchai, 2024). The CTGAN-ENN method is a baseline of this study on data-level solutions.

Algorithm-level solution

CSL as an algorithm-level solution has been proposed in several research to enhance classical machine learning algorithm performance, comparing false-positive and false-negative as a cost of learning on the classifier. The experiments aim to check the total cost of misclassification (Kumaravel & Vijayan, 2023). The lower cost indicates better performance of the classifier with the specific value of cost ratio. Other approaches proposed a comparison of class-dependent and instance-dependent as a type of cost in CSL, based on their results instance-dependent cost-sensitive learning achieved better performance than class-dependent cost-sensitive learning (Vanderschueren et al., 2022). Several studies applied CSL to specific machine learning algorithms. A constructive procedure to extend the standard support vector machine (SVM) loss function to optimize the classifier for class imbalance or class-specific costs (Iranmehr, Masnadi-Shirazi & Vasconcelos, 2019). Self-adaptive cost weights based on SVM cost-sensitive ensemble for imbalanced data classification (Tao et al., 2019). The results of their experiments showed improved generalization performance of the SVM. Decision tree (DT) classifier with cost-sensitive learning was proposed to handle the data imbalanced problem as well, and the result was shown to enhance the recognition of the minority class (Krawczyk, Woźniak & Schaefer, 2014). Another approach to DT proposed the partially observable Markov decision processes (POMDP) on cost-sensitive modeling in the DT algorithm. Their method proved to be more effective for a variety of misclassification costs (Maliah & Shani, 2021).

Hybrid solution

The main problem with data-level hybrid sampling methods is that the data often becomes imbalanced again after reducing overlap through under-sampling. This leads classical machine learning models to naturally bias towards the majority class. To address this issue, this study proposes an integrated framework combining both data-level and algorithm-level solutions. A GAN-based hybrid sampling method is used in the data preprocessing phase due to its capabilities, which surpass those of traditional sampling methods. The algorithm-level solution is addressed through cost-sensitive learning. Recent studies show that cost-sensitive learning in classical machine learning results in satisfactory performance (Adiputra & Wanchai, 2024). Classical machine learning models offer efficient processing times, making them highly useful for customer churn prediction tasks, where timely and accurate insights are essential. Combining a data-level solution (CTGAN-ENN) with a cost-sensitive learning approach can significantly improve classification performance for the minority class, ensuring that both synthetic data and cost penalties guide the model to prioritize minority-class learning.

Materials and Methods

CostLearnGAN framework

The entire process of the CostLearnGAN framework is shown in Fig. 1. CTGAN-ENN enhances the customer churn prediction by addressing class imbalance and overlaps through a two-step process: oversampling with CTGAN and undersampling with ENN. First, CTGAN generates realistic synthetic samples for the minority class, capturing complex data distributions and learning from the original dataset. CTGAN handles categorical and numerical features by selecting real data points during training and conditions the generator on them. CTGAN handles discrete features by conditional sampling during training. CTGAN learns to condition discrete variables, improving their representation in the generated data (Xu et al., 2019). The generator takes these learned patterns and attempts to produce synthetic samples that follow the same structure and statistical distribution as the real dataset. This means the model learns patterns directly from the dataset rather than relying on random noise as input. CTGAN follows the standard Wasserstein GAN with the gradient penalty (WGAN-GP) framework.

(1) LG=−Ex^∼Pg[D(x^)]

(2) LD=Ex^∼Pg[D(x^)]−Ex∼Pr[D(x)]+λEx~∼Px~[(|∇x~D(x~)|2−1)2]

Figure 1 CostLearnGAN framework.

Formula (1) is the generator loss LG, designed to fool the discriminator by maximizing the expected output of the discriminator on generated samples, effectively pushing it to generate realistic data. Formula (2) is the discriminator loss LD computes the Wasserstein distance between real and generated data distributions, encouraging better separability between them. To enforce the Lipschitz constraint, a gradient penalty term is added, which penalizes deviations of the gradient norm from 1, ensuring more stable and robust training.

The generated data is then concatenated with the original dataset to create a more balanced representation. However, oversampling can produce overlapping samples, which are mitigated by ENN. ENN removes misclassified and ambiguous samples by evaluating their nearest neighbors, ensuring a cleaner and more refined dataset (Adiputra & Wanchai, 2024). This hybrid approach improves classifier robustness, enhances generalization, and ultimately boosts the performance of customer churn. These final datasets from the data-level method, were then subjected to a five-fold cross-validation process to ensure robustness and reliability in model evaluation. Cross-validation splits the data into five subsets, where each subset is used as a validation set while the remaining four subsets are used for training. This process is repeated five times, with each subset serving as the validation set once. The validated datasets are then utilized to train three different machine learning classifiers. These classifiers are evaluated by varying class weights to address any residual imbalance and optimize model performance. The main function of cost-sensitive learning is fine-tuning the hyperparameter of the classifier in class weight hyperparameter. This involves adjusting the weights assigned to the classes in the loss function to handle class imbalance effectively. The class weights are adjusted in a range of values {10, 20, 30, 40, 50, 60, 70, 80, 90, 100} where x represents the weight assigned to the non-churn class (0) and y represents the weight assigned to the churn class (1). This framework aims to improve the prediction accuracy of customer churn by effectively handling class imbalance through synthetic data generation and model evaluation with optimized class weight. That hyperparameter plays an important role in handling imbalanced datasets. In default machine learning algorithms tend to be learned in the non-churn class because it’s a majority data and has a poor performance in the churn class. By using cost-sensitive learning the model will be more likely to learn the characteristics of the churn class, leading to better prediction performance when applying correct weighting to the hyperparameter.

Evaluation metrics

This study used three main metrics to evaluate the performance of CostLearnGAN on classical machine learning, the evaluation metrics are AUC-ROC, F1-score, and geometric mean (G-mean). Calculating the F1-score and G-mean requires the values of precision (PR), recall (RC), and specificity (SP), which are measured using Formulas (3)–(5) as follows:

(3) PR=TP/(TP+FP)

(4) RC=TP/(TP+FN)

(5) SP=TN/(TN+FP)

True positive (TP) refers to the number of churn customers predicted as churn, and true negative (TF) refers to a total of non-churn customers predicted not churn. False positive (FP) is the number of non-churn customers who are incorrectly predicted as churn. False negative (FN) is the number of churn customers who are incorrectly predicted as not churn.

(6) F1=2∗((PR∗RC))/((PR+RC))

(7) G−mean=√(RC∗SP)

F1-score considers both the precision (the accuracy of the positive predictions) and the recall (how many of the actual positives the model captures), calculated by Formula (6). G-Mean, commonly applied to imbalanced datasets, represents the geometric mean of sensitivity and specificity, offering a balanced performance metric. It is calculated using Formula (7).

Experiments

Datasets

Table 1 provides a comprehensive overview of various datasets utilized in the analysis, detailing the number of features, total data points, and the imbalance ratio of datasets. The datasets span different domains, including banking, mobile services, telecommunications, and insurance. All datasets in this study were collected from the Kaggle platform.

Table 1 Datasets overview.

Dataset	Features	Data number	Imbalance ratio	
Bank	13	10,000	3.9	
Mobile	65	66,469	3.7	
Telco 1	19	7,044	2.7	
Telco 2	19	4,250	6.1	
Telco 3	15	3,150	5.3	
Insurance	16	33,909	7.5	

The Bank dataset comprises 13 features and 10,000 data points, with an imbalance ratio of 3.9. The Mobile dataset is significantly larger, featuring 65 attributes and 66,469 data points, with an imbalance ratio of 3.7. The Telco 1 dataset includes 19 features and 7,044 data points, maintaining an imbalance ratio of 2.7, while Telco 2, also with 19 features, consists of 4,250 data points and exhibits a higher imbalance ratio of 6.1. The Telco 3 dataset contains 15 features across 3,150 data points, presenting an imbalance ratio of 5.3. Lastly, the Insurance dataset is characterized by 16 features and 33,909 data points, with the highest imbalance ratio of 7.5 among the datasets listed at 4.2

Experimental settings

In this study, three classical machine learning algorithms DT, logistic regression (LR), and SVM along with two boosting algorithms, Random Forest (RF) and LightGBM (LGBM), are used. The experimental procedure involves performing 5-fold cross-validation three times for each algorithm to ensure robust and reliable results. This approach mitigates the impact of any potential variability or biases in the data, providing a more comprehensive evaluation of the model performance.

The preprocessing process begins with data acquisition, where datasets are open public that are collected from the Kaggle platform. To ensure data quality, data cleaning is performed by handling missing values using mean imputation for numerical features and mode imputation for categorical features, removing duplicates, encoding categorical variables, and scaling numerical features if necessary. After cleaning, the CTGAN-ENN method is applied to address the class imbalance. CTGAN generates synthetic samples for the minority class, capturing complex data distributions. The synthetic data is then concatenated with the original dataset, and ENN is used to remove overlapping samples, ensuring a high-quality, balanced dataset. Notably, this study does not apply feature selection, allowing models to learn from the full set of customer attributes without introducing selection bias. Once the final dataset is prepared, cost-sensitive learning is implemented, as detailed in Algorithm 1, with Optuna tuning class weights to further enhance model performance by effectively addressing class imbalance at the algorithm level.

Algorithm 1 Cost-sensitive fine tuning.

  Input: Final Dataset after Sampling Method	
  Output: Best Performance of Algorithm	
  1 Define x = {10, 20, 30, 40, 50, 60, 70, 80, 90, 100}	
  2 Define y = {10, 20, 30, 40, 50, 60, 70, 80, 90, 100}	
  3 For each classifier:	
  4   For each combination of (x, y):	
  5     Set classifier class weight {0:x, 1: y}	
  6     Train classifier on dataset	
  7     Evaluate performance and store results	
  8   End for	
  9 End for	
  10 Identify the best (x, y) combination based on evaluation metrics	
  11 Define the best classification result	

The cost-sensitive fine-tuning of the sampling method outlines the process for identifying the optimal performance of classifiers using a final dataset that has undergone processing with the sampling methods. The algorithm evaluates classifiers based on various class weight configurations to determine the best classification result. The steps are as follows: 1. Input and output specification: The algorithm begins with the final dataset obtained after applying sampling methods. The goal is to determine the best performance of each classifier by adjusting the optimal value of class weight.

2. Parameter initialization: Two sets of class weight values, x and y, are initialized. Both sets contain the values {10,20,30,40,50,60,70,80,90,100}. x is the class weight value for the non-churn class, while y is the class weight value for the churn class.

3. Algorithm setting robust to not churn class: I. For each classifier, the algorithm iterates over the set y.

II. Within each iteration, the classifier’s class weight is set to {0:100,1:y}.

III. The classifier is then evaluated 10 times, and the best result from these evaluations is recorded as x100.

4. Algorithm setting robust to churn class: I. Similarly, the algorithm iterates over the set x for each classifier.

II. In each iteration, the classifier’s class weight is set to {0:x,1:100}.

III. The classifier undergoes 10 evaluations, and the best result from these evaluations is recorded as y100.

5. Comparison and selection: I. After evaluating the classifiers with both sets of class weights, the algorithm compares the results obtained from x100 and y100.

II. Based on this comparison, the algorithm defines the best classification result for each classifier.

The experiment treated all sampling methods under the same condition by Algorithm 1, to provide an unbiased, systematic comparison of the sampling methods under investigation and ensure that all sampling methods are subjected to the same treatment conditions. In Python code, the hyperparameter value is selected using the optuna library. Optuna was chosen because of its efficient and automated hyperparameter optimization framework that leverages Bayesian optimization to find the best parameter values (Akiba et al., 2019). Optuna is useful for tuning class weights, as balancing the trade-off between majority and minority classes is crucial for improving model performance in imbalanced datasets. Optuna iteratively adjusted the hyperparameters based on previous evaluations, efficiently narrowing down the optimal values that maximized model performance based on the F1 metric. By comparing different trials, the algorithm will choose the best-performing set of hyperparameters. The difference in weight values for each class affects classification performance because it directly influences the model’s decision boundary and its sensitivity to different classes.

Results

Tables 2–4 presents the results for each dataset across various classifiers, evaluated in terms of F1-score, AUC, and G-mean. There are 18 different results based on the experiments with each metric. The experiment compared the proposed method with six different conventional sampling methods. The best parameter is the class weight value which has the best performance on the proposed method.

Table 2 Experimental comparison on F1-score.

Alg	Dataset	NONE	SM	SE	WG	WE	CT	Proposed	
DT	Bank	0.684	0.800	0.812	0.966	0.967	0.782	0.902	
	Mobile	0.768	0.862	0.932	0.941	0.967	0.894	0.993	
	Telco1	0.658	0.792	0.937	0.922	0.965	0.800	0.944	
	Telco2	0.841	0.875	0.883	0.985	0.985	0.926	0.968	
	Telco3	0.887	0.951	0.978	0.956	0.856	0.957	0.986	
	Insurance	0.701	0.900	0.944	0.901	0.846	0.926	0.978	
LR	Bank	0.611	0.670	0.678	0.637	0.705	0.651	0.721	
	Mobile	0.799	0.820	0.869	0.878	0.877	0.903	0.915	
	Telco1	0.737	0.793	0.926	0.801	0.849	0.808	0.857	
	Telco2	0.640	0.704	0.696	0.881	0.877	0.836	0.940	
	Telco3	0.718	0.793	0.853	0.731	0.693	0.757	0.852	
	Insurance	0.721	0.836	0.884	0.872	0.877	0.835	0.901	
SVM	Bank	0.480	0.555	0.634	0.811	0.796	0.629	0.841	
	Mobile	0.798	0.824	0.898	0.869	0.873	0.912	0.943	
	Telco1	0.707	0.738	0.899	0.796	0.844	0.802	0.901	
	Telco2	0.758	0.822	0.832	0.928	0.931	0.918	0.977	
	Telco3	0.657	0.772	0.858	0.683	0.673	0.800	0.929	
	Insurance	0.755	0.884	0.935	0.831	0.876	0.923	0.972	
RF	Bank	0.573	0.856	0.888	0.787	0.845	0.825	0.886	
	Mobile	0.685	0.901	0.956	0.684	0.856	0.918	0.993	
	Telco1	0.565	0.849	0.958	0.847	0.955	0.838	0.927	
	Telco2	0.830	0.937	0.963	0.969	0.973	0.970	0.995	
	Telco3	0.841	0.971	0.985	0.726	0.827	0.967	0.986	
	Insurance	0.525	0.939	0.972	0.837	0.923	0.931	0.958	
LGBM	Bank	0.618	0.853	0.895	0.618	0.711	0.828	0.890	
	Mobile	0.708	0.889	0.944	0.712	0.853	0.903	0.989	
	Telco1	0.571	0.848	0.962	0.855	0.951	0.845	0.931	
	Telco2	0.851	0.927	0.954	0.974	0.976	0.974	0.995	
	Telco3	0.881	0.977	0.990	0.785	0.829	0.970	0.984	
	Insurance	0.624	0.937	0.972	0.873	0.945	0.927	0.965	
Note:

Bold value indicates the best performance of F1-score on experiments.

Table 3 Experimental comparison on AUC.

Alg	Dataset	NONE	SM	SE	WG	WE	CT	Proposed	
DT	Bank	0.689	0.800	0.814	0.967	0.967	0.781	0.906	
	Mobile	0.778	0.846	0.933	0.970	0.969	0.894	0.993	
	Telco1	0.662	0.793	0.937	0.925	0.965	0.800	0.944	
	Telco2	0.843	0.875	0.877	0.985	0.986	0.926	0.967	
	Telco3	0.904	0.955	0.977	0.972	0.858	0.961	0.985	
	Insurance	0.711	0.900	0.943	0.908	0.847	0.926	0.977	
LR	Bank	0.712	0.741	0.769	0.790	0.702	0.775	0.847	
	Mobile	0.879	0.884	0.936	0.946	0.933	0.934	0.946	
	Telco1	0.841	0.879	0.979	0.886	0.928	0.899	0.941	
	Telco2	0.758	0.704	0.768	0.927	0.930	0.904	0.984	
	Telco3	0.879	0.873	0.928	0.906	0.829	0.863	0.932	
	Insurance	0.875	0.902	0.949	0.916	0.912	0.911	0.962	
SVM	Bank	0.626	0.714	0.697	0.839	0.845	0.724	0.928	
	Mobile	0.883	0.889	0.957	0.937	0.950	0.968	0.957	
	Telco1	0.807	0.817	0.964	0.873	0.924	0.894	0.956	
	Telco2	0.834	0.822	0.918	0.961	0.964	0.964	0.997	
	Telco3	0.915	0.902	0.913	0.954	0.871	0.905	0.976	
	Insurance	0.906	0.944	0.981	0.925	0.910	0.974	0.995	
RF	Bank	0.852	0.929	0.942	0.938	0.961	0.910	0.973	
	Mobile	0.900	0.952	0.993	0.939	0.974	0.971	0.999	
	Telco1	0.828	0.927	0.989	0.931	0.993	0.920	0.988	
	Telco2	0.913	0.983	0.988	0.986	0.988	0.986	0.999	
	Telco3	0.983	0.992	0.999	0.940	0.946	0.995	0.998	
	Insurance	0.925	0.987	0.995	0.990	0.998	0.985	0.996	
LGBM	Bank	0.861	0.928	0.951	0.918	0.947	0.919	0.977	
	Mobile	0.909	0.952	0.990	0.947	0.977	0.975	0.996	
	Telco1	0.837	0.928	0.991	0.855	0.994	0.934	0.993	
	Telco2	0.908	0.977	0.984	0.985	0.987	0.987	0.999	
	Telco3	0.988	0.996	0.999	0.944	0.955	0.996	0.998	
	Insurance	0.934	0.988	0.996	0.994	0.998	0.989	0.997	
Note:

Bold value indicates the best performance of AUC on experiments.

Table 4 Experimental comparison on G-mean.

Alg	Dataset	NONE	SM	SE	WG	WE	CT	Proposed	
DT	Bank	0.671	0.800	0.811	0.966	0.967	0.781	0.905	
	Mobile	0.785	0.862	0.932	0.945	0.967	0.894	0.993	
	Telco1	0.649	0.793	0.936	0.922	0.965	0.799	0.946	
	Telco2	0.843	0.875	0.876	0.986	0.985	0.926	0.967	
	Telco3	0.890	0.950	0.977	0.963	0.855	0.957	0.986	
	Insurance	0.681	0.900	0.943	0.900	0.844	0.926	0.977	
LR	Bank	0.645	0.670	0.662	0.680	0.708	0.650	0.737	
	Mobile	0.817	0.820	0.866	0.866	0.878	0.903	0.920	
	Telco1	0.759	0.793	0.926	0.803	0.846	0.808	0.870	
	Telco2	0.692	0.704	0.697	0.858	0.859	0.831	0.936	
	Telco3	0.799	0.791	0.851	0.833	0.763	0.756	0.849	
	Insurance	0.799	0.836	0.883	0.858	0.851	0.834	0.902	
SVM	Bank	0.556	0.541	0.607	0.772	0.771	0.628	0.838	
	Mobile	0.825	0.823	0.898	0.880	0.878	0.911	0.936	
	Telco1	0.727	0.737	0.901	0.794	0.834	0.801	0.899	
	Telco2	0.739	0.820	0.843	0.920	0.924	0.917	0.976	
	Telco3	0.747	0.765	0.854	0.812	0.738	0.799	0.925	
	Insurance	0.831	0.883	0.934	0.832	0.846	0.922	0.971	
RF	Bank	0.662	0.858	0.888	0.787	0.845	0.828	0.886	
	Mobile	0.769	0.870	0.936	0.793	0.875	0.895	0.971	
	Telco1	0.671	0.846	0.952	0.853	0.963	0.839	0.941	
	Telco2	0.860	0.940	0.951	0.968	0.974	0.972	0.995	
	Telco3	0.889	0.966	0.983	0.812	0.858	0.965	0.986	
	Insurance	0.653	0.937	0.968	0.893	0.944	0.932	0.965	
LGBM	Bank	0.723	0.851	0.878	0.771	0.825	0.836	0.918	
	Mobile	0.779	0.861	0.920	0.812	0.857	0.861	0.961	
	Telco1	0.683	0.846	0.956	0.857	0.956	0.848	0.938	
	Telco2	0.878	0.928	0.944	0.974	0.975	0.974	0.996	
	Telco3	0.940	0.974	0.989	0.848	0.863	0.968	0.983	
	Insurance	0.815	0.938	0.970	0.935	0.959	0.929	0.967	
Note:

Bold value indicates the best performance of G-mean on experiments.

CostLearnGAN achieved the best result on 19 out of 30 scenarios in terms of F1-score as shown in Table 2. Indicates that the proposed method outperforms compared to another sampling method on harmonic means of precision and recall. CostLearnGAN obtains the best result on 20 out of 30 scenarios in terms of AUC-ROC as shown in Table 3. CostLearnGAN consistently produced the highest or near-highest AUC values across the majority of datasets and algorithms, demonstrating its versatility and effectiveness in comparison to traditional sampling methods.

CostLearnGAN was optimal on G-mean metrics as Table 4 shows, the framework outperforms on 19 out of 30 scenarios. CostLearnGAN consistently yielded the highest G-Mean scores across all datasets and algorithms, highlighting its strength in addressing class imbalance while maintaining good classifier performance for both majority and minority classes.

Table 5 provides the mean rank score of all algorithms and sampling methods across all metrics. Mean rank scores indicate the relative performance of each sampling method, with lower scores representing better ranks. CostLearnGAN consistently shows the lowest mean rank scores, indicating superior performance. Specifically, CostLearnGAN achieves the best scores across all metrics for DT, LR, SVM, and RF algorithms. Although the SE method occasionally matches the proposed method’s performance for LGBM, the proposed approach generally dominates, indicating its effectiveness in yielding higher predictive accuracy compared to alternative methods like no sampling method (NONE), SMOTE (SM), SMOTE + ENN (SE), Wasserstein GAN + Gradient Penalty (WGAN-GP) (WG), WG + ENN (WE), and Conditional Tabular GAN (CT) methods.

Table 5 Mean rank score.

Alg	Metric	NONE	SM	SE	WG	WE	CT	Proposed	
DT	F1-score	6.83	5.50	3.33	3.08	3.08	4.33	1.83	
	AUC	6.83	5.50	3.33	2.75	3.25	4.50	1.83	
	G-mean	6.83	5.42	3.33	2.92	3.17	4.50	1.83	
LR	F1-score	6.83	4.83	3.00	4.17	3.67	4.17	1.33	
	AUC	6.17	5.83	2.83	2.92	4.67	4.33	1.25	
	G-mean	6.50	5.17	3.25	3.58	3.33	4.83	1.33	
SVM	F1-score	7.00	5.33	3.00	4.33	3.83	3.50	1.00	
	AUC	6.17	5.67	3.42	4.00	4.08	3.25	1.42	
	G-mean	6.50	5.83	3.00	3.83	4.17	3.50	1.17	
RF	F1-score	6.50	3.83	2.00	5.83	4.00	4.16	1.66	
	AUC	6.66	5.00	2.16	4.83	2.50	4.83	1.66	
	G-mean	6.66	4.33	2.16	5.50	3.33	4.33	1.66	
LGBM	F1-score	6.50	4.16	1.83	5.33	3.83	4.16	1.83	
	AUC	6.66	4.83	2.66	5.50	2.66	3.83	1.50	
	G-mean	6.66	4.16	2.00	5.16	3.66	4.16	1.66	
Note:

Bold value indicates the best mean rank score in each evaluation metric of experiments.

To strengthen the analysis Table 6 provides an ANOVA statistics test followed by a post-hoc test Tukey’s honestly significant difference (HSD) test with = 0.05. Based on the post-hoc test CostLearnGAN significantly outperforms other sampling methods in terms of mean rank scores, as indicated by the large mean differences, the significant p-values, and the confidence intervals that do not include zero.

Table 6 ANOVA post-hoc analysis.

Group 1	Group 2	Mean diff	p-value	Lower	Upper	Significant	
CT	Proposed	−2.628	0	−3.2931	−1.9629	TRUE	
NONE	Proposed	−5.0893	0	−5.7545	−4.4242	TRUE	
Proposed	SE	1.2227	0	0.5575	1.8878	TRUE	
Proposed	SM	3.4953	0	2.8302	4.1605	TRUE	
Proposed	WE	2.018	0	1.3529	2.6831	TRUE	
Proposed	WG	2.718	0	2.0529	3.3831	TRUE	

Figures 2–7 provided a comprehensive analysis of the effect of fine-tuning hyperparameters on each dataset with one of the best machine learning algorithm results based on the F1-score metric. The visualization is represented by scatter plots, where red and blue dots indicate two different classes. The first column shows the original dataset without any decision boundary applied, illustrating the distribution of the two classes in the feature space. The rows indicate three different types of data clusters: moon shape, circle shape, and linear separated shape. The color gradients indicate the decision regions, where red and blue correspond to different classes. The results highlight how adjusting class weights impacts decision boundaries, making the model more responsive to class imbalance.

Figure 2 Effect of class weight tuning on LR decision boundaries for the Bank dataset based on F1-score metric.

Figure 3 Effect of class weight tuning on DT decision boundaries for the Mobile dataset based on F1-score metric.

Figure 4 Effect of class weight tuning on SVM decision boundaries for the Telco 1 dataset based on F1-score metric.

Figure 5 Effect of class weight tuning on DT decision boundaries for the Telco 2 dataset based on F1-score metric.

Figure 6 Effect of class weight tuning on SVM decision boundaries for the Telco 3 dataset based on F1-score metric.

Figure 7 Effect of class weight tuning on SVM decision boundaries for the Insurance dataset based on F1-score metric.

The color scale in the decision boundary plots represents the classifier’s prediction confidence for each class. Red regions indicate areas predicted as the notchurn class, while blue regions indicate areas predicted as the churn class. Darker shades correspond to higher confidence in the respective class prediction, whereas lighter shades represent lower confidence. The white or neutral zones near the class boundaries signify regions of uncertainty where the classifier is less confident. Contour labels such as 0.80, 0.95, and 1.00 denote specific confidence levels, illustrating how strongly the classifier predicts a given class in those regions.

The best results for the bank dataset were achieved using the LR algorithm, as demonstrated in Fig. 2. The cost-sensitive learning method exhibited superior classification performance for data with circular shapes. Figure 3 shows that the DT algorithm outperformed all other methods for various cluster types in the mobile dataset. For the Telco 1 dataset, SVM showed superior performance in moon-shaped and linearly separable clusters, as depicted in Fig. 4.

The Telco 2 dataset achieved the best results using the DT algorithm, effectively handling moon-shaped clusters as shown in Fig. 5. Interestingly, the SVM algorithm performed well on both the Telco 3 and insurance datasets, particularly with moon-shaped clusters as shown in Figs. 6 and 7. Across the experiments, SVM outperformed other algorithms, especially when combined with cost-sensitive learning, as evidenced by the F1-score metric. SVM achieved the best performance across the evaluated datasets, demonstrating superior accuracy and robustness in handling the classification tasks. Given its exceptional performance, this study provided the total cost associated with SVM, by multiplying the weight of cost-sensitive learning with the confusion matrix values. This approach allows us to quantify the real impact of misclassification errors in terms of cost (Zelenkov, 2019). The best class weight hyperparameter can be seen in Table 7, the confusion matrix shown in Table 8 and the measurement of total cost represents in Table 9. These results highlight the superiority of cost-sensitive methods on WE and proposed methods in reducing overall costs compared to other approaches. Moreover, the type I and type II errors or false positives and false negatives also can be seen in Table 8. CostLearnGAN demonstrates significant improvements over baseline methods such as NONE and SMOTE, consistently outperforming them in most cases. It effectively balances false positives and false negatives, leading to more robust predictions compared to traditional oversampling techniques. It performs exceptionally well on the Bank dataset, achieving the highest recall and true positive count, making it particularly effective in identifying churners. However, its performance varies across datasets, struggling in certain Telco cases (Telco 2 and Telco 3) where it predicts too few true positives. In these instances, the model tends to prioritize reducing false positives, which inadvertently leads to a higher number of missed churners (false negatives) and lower recall.

Table 7 Best class weight hyperparameter.

Dataset		NONE	SM	SE	WG	WE	CT	Proposed	
		0	1	0	1	0	1	0	1	0	1	0	1	0	1	
Bank	0	0	20	0	100	0	20	0	70	0	100	0	100	0	100	
	1	100	0	90	0	100	0	100	0	100	0	80	0	100	0	
Telco 1	0	0	20	0	20	0	100	0	100	0	100	0	80	0	40	
	1	100	0	100	0	70	0	50	0	10	0	100	0	100	0	
Telco 2	0	0	90	0	70	0	100	0	100	0	100	0	100	0	100	
	1	100	0	100	0	50	0	40	0	60	0	60	0	20	0	
Telco 3	0	0	60	0	80	0	90	0	70	0	30	0	20	0	100	
	1	100	0	100	0	100	0	100	0	100	0	100	0	100	0	
Mobile	0	0	100	0	100	0	100	0	50	0	30	0	100	0	60	
	1	70	0	80	0	70	0	100	0	100	0	100	0	100	0	
Insurance	0	0	100	0	100	0	100	0	100	0	100	0	80	0	100	
	1	100	0	60	0	100	0	20	0	20	0	100	0	100	0	

Table 8 Confusion matrix on the SVM algorithm.

	
Actual	Predicted	
	NONE	SM	SE	WG	WE	CT	Proposed	
Dataset	0	1	0	1	0	1	0	1	0	1	0	1	0	1	
Bank	0	619	974	665	928	0	574	2,549	0	1,137	4	1,302	291	1,526	67	
	1	107	300	433	1,160	0	714	650	1	648	3	780	813	173	363	
Telco 1	0	534	501	534	501	475	53	1,024	151	605	1	729	306	885	150	
	1	37	337	37	337	73	568	357	577	257	677	201	834	84	465	
Telco 2	0	727	3	577	154	388	38	727	3	528	2	731	0	731	0	
	1	81	39	116	614	189	515	115	605	66	654	133	597	15	602	
Telco 3	0	503	28	318	213	281	114	461	308	314	138	254	277	505	26	
	1	64	35	7	524	35	424	64	397	11	450	1	530	86	274	
Mobile	0	9,841	672	9,033	1,480	7,034	288	16,866	1,135	15,093	498	9,307	1,206	9,307	708	
	1	2,219	1,691	2,219	8,293	1,399	5,819	774	2219	515	2,478	1,271	9,241	513	6,511	
Insurance	0	5,859	130	5,328	661	4,512	387	5,986	6	5,171	16	5,325	664	5,744	244	
	1	544	249	785	5,203	285	5,576	780	5,210	567	5,423	517	5,471	192	3,750	

Table 9 Total cost analysis.

Dataset/Method	NONE	SM	SE	WG	WE	CT	Proposed	
Bank	30,180	131,770	11,480	65,000	65,200	91,500	24,000	
Telco 1	13,720	13,720	10,410	31,950	2,670	44,580	14,400	
Telco 2	8,370	22,380	13,250	4,900	4,160	7,980	300	
Telco 3	8,080	17,740	13,760	27,960	5,240	5,640	11,200	
Mobile	169,480	325,520	126,730	134,150	66,440	247,700	93,780	
Insurance	67,400	113,200	67,200	16,200	12,940	104,820	43,600	
Average	49,538	104,055	40,472	46,693	26,108	83,703	31,213	
Note:

Bold value represents the smallest total cost, underscore value represents the second smallest total cost.

To verify the robustness of each algorithm (Tao et al., 2019), this study provided the measurement of algorithm robustness which is shown in Formulas (8)–(10).

(8) Am=MAucmmaxm⁡Aucm

(9) Fm=MFMeasuremmaxm⁡FMeasurem

(10) Gm=MGMeanmmaxm⁡GMeanm

Figures 8–10 show the comparison of robustness results, performance of the proposed method in terms of robustness across multiple datasets compared to the other techniques, highlighting its effectiveness and reliability. MAucm is the mean of AUC-ROC (Fig. 8), MFMeasurem is the mean of F1-score (Fig. 9), andMGMeanm is the mean of G-Mean (Fig. 10), for m algorithm with different datasets. The larger value of Am, Fm, and Gm indicates better robustness of the algorithm. The proposed method of this study consistently achieved the highest value on algorithm robustness indicating that CostLearnGAN is effectively applicable for imbalanced and overlapped data in customer churn prediction tasks.

Figure 8 Comparison of algorithm robustness in term of AUC.

Figure 9 Comparison of algorithm robustness in term of F1-score.

Figure 10 Comparison of algorithm robustness in term of G-mean.

Discussion

CostLearnGAN resulted in substantial improvements across both AUC and F1-score, highlighting its effectiveness in handling complex, imbalanced data. The optimal parameters for each configuration indicate that tuning the cost and weight parameters is crucial for maximizing performance. Robustness measurement proved that the proposed method is highly effective in enhancing the performance of customer churn prediction task, followed by traditional methods (WE), but the performance of WE varied depending on the dataset and algorithm. The Bank dataset gained the lowest performance on the proposed method compared to other datasets, this might happen because the sparsity or overlap data was not reduced well on the ENN method. The limitation of this study is fine-tuning can be highly dependent on the underlying data distribution. Additionally, the process of under-sampling probably removes the important real data points and potentially leads to information loss. Modifying the algorithm that ensures real data points are not affected by the under-sampling process and implementing an adaptive learning technique that adjusts model parameters dynamically based on shifts in data distribution might be useful for the development of future frameworks.

The results of this study demonstrate superior performance compared to previous studies using the same dataset, as shown in Table 8. Except for the Bank dataset on F1-score, CostLearnGAN consistently outperforms prior approaches in both F1-score and AUC. This highlights the effectiveness of CostLearnGAN in improving customer churn classification performance.

Unlike traditional oversampling techniques such as SMOTE, which rely on interpolation and often fail to capture the complex data distribution, or GBHS (Zhu et al., 2022), which is not extended to the algorithm level. The incorporation of CTGAN-ENN and cost-sensitive learning fine-tunes class weights dynamically, allowing the model to better adapt to varying levels of imbalance across datasets. This dual-level optimization enhancing data quality with CTGAN-ENN while simultaneously improving classifier robustness through cost-sensitive learning makes our approach more effective in handling imbalanced datasets compared to previous methods.

This study’s results align with findings from GAN based hybrid method (Ding et al., 2023; Zhu et al., 2022) and cost-sensitive learning method (Tao et al., 2019; Vanderschueren et al., 2022; Zhou et al., 2023). This study demonstrates decent performance over recent works across the same datasets analyzed. However, an exception is observed with the bank dataset, where a voting classifier outperforms the proposed method. This specific detail is illustrated in Table 10. Integrating the hybrid sampling method and hyperparameter optimization strategies substantially improves model efficacy in customer churn prediction tasks. The proposed method works on different types of imbalanced ratios and in different kinds of machine learning algorithms.

Table 10 Comparison results with latest customer churn prediction studies.

Author	Dataset	Sampling method	Algorithm	F1-score	AUC	
Zhu et al. (2022)	Telco	GBHS, WGAN-GP based	Gradient boosting machine	–	0.660	
Wen et al. (2022)	Telco	None	DCN + ASL	0.630	0.840	
	Insurance			0.620	0.930	
Gowd et al. (2023)	Telco	None	Random forest	0.800	0.840	
Poudel, Pokharel & Timilsina (2024)	Telco	None	Gradient boosting machine	0.600	0.860	
Bhuria et al. (2025)	Bank	None	Voting classifier	0.900	–	
This study	Telco	CTGAN + ENN	SVM + CSL	0.901	0.956	
	Bank			0.841	0.928	
	Insurance			0.972	0.995	
Note:

Bold value represents the highest score achieved.

To extend the usability of CostLearnGAN, further research is needed to determine its feasibility for real-time applications. Investigating ways to optimize training and inference speed would be a valuable direction for future work. CostLearnGAN also has the potential to be extended beyond customer churn prediction. Exploring its application in domains such as fraud detection, credit score, or financial risk assessment.

Conclusions

In this article, a hybrid sampling method with a combination of algorithm-level solutions was proposed for imbalanced and overlapped customer churn prediction data. This study proposed a cost-sensitive learning approach called CostLearnGAN in order to improve the performance of classical machine learning algorithms. Cost-sensitive learning using class weight hyperparameter optimization was tried to solve the problem. Experimental results show CostLearnGAN leads the mean rank of AUC-ROC, F1-score, and G-mean evaluation metrics, the result also shows CostLearnGAN was more robust than another sampling method in all classical machine learning algorithms.

Customer churn prediction data has a specific problem with imbalanced and overlapped data. CostLearnGAN addresses the problem and enhances customer churn prediction by integrating GAN-based data augmentation and cleaning the overlap by the under-sampling method, extended with cost-sensitive learning at both the data and algorithm levels. Unlike traditional oversampling methods like SMOTE, which rely on interpolation, CostLearnGAN generates high-quality synthetic samples that better reflect real data distributions, improving classifier robustness. Additionally, the framework optimizes class weights on the algorithm level to enhance generalization across different classifiers and datasets. Through extensive empirical evaluation, this study demonstrates that CostLearnGAN outperforms existing techniques in predictive accuracy and recall. By bridging data-level and algorithm-level approaches, this research provides a holistic solution to tackle class imbalance, ensuring both improved data quality and optimized model performance for customer churn prediction.

The limitation of this study is, that ENN works naturally so there is a possibility ENN method deletes a real instance, not the synthetic one. Modifying ENN might be explored as one aspect of potential improvements. CTGAN might cause anomalies in data that are not detected by ENN. For further research, modify the under-sampling technique to work only on synthetic data. Another possible improvement is on the GAN part, modifying the generator and discriminator of CTGAN by using a transformer-based based might promising rather than the default neural network setting. On the algorithm level reinforcement learning for hyperparameter tuning might allow the model to learn from real-time data feedback, allowing cost-sensitive learning work more dynamically than just a static weight value.

Supplemental Information

Supplemental Information 1 Dataset dictionary and codebook.

Supplemental Information 2 Python code of cost-sensitive learning in all compared methods.

Supplemental Information 3 Customer churn datasets.

This study was conducted in collaboration between AIDA Lab in College of Computing and Rebecca Lab, Feng Chia University, Taiwan.

Abbreviations

CSL Cost-Sensitive Learning

CT Conditional Tabular GAN

DT Decision Tree

ENN Edited Nearest Neighbor

RF Random Forest

GAN Generative Adversarial Network

LGBM Light Gradient-Boosting Machine

LR Logistic Regression

NONE No Sampling Method

SVM Support Vector Machine

SM Synthetic Minority Oversampling Technique (SMOTE)

SE SMOTE + ENN

WG Wasserstein GAN + Gradient Penalty (WGAN-GP)

WE WG + ENN

Additional Information and Declarations

Competing Interests

The authors declare there are no competing interests.

Author Contributions

I Nyoman Mahayasa Adiputra conceived and designed the experiments, performed the experiments, analyzed the data, performed the computation work, prepared figures and/or tables, authored or reviewed drafts of the article, and approved the final draft.

Paweena Wanchai conceived and designed the experiments, performed the experiments, analyzed the data, performed the computation work, authored or reviewed drafts of the article, and approved the final draft.

Pei-Chun Lin performed the experiments, authored or reviewed drafts of the article, and approved the final draft.

Data Availability

The following information was supplied regarding data availability:

The datasets of this study are available at Kaggle:

- Bank customer churn dataset, https://www.kaggle.com/datasets/shrutimechlearn/churn-modelling

- Mobile customer churn dataset, https://www.kaggle.com/datasets/undersc0re/predict-the-churn-risk-rate

- IBM telecommunication customer churn dataset (Telco1), https://www.kaggle.com/datasets/blastchar/telco-customer-churn

- Orange telecommunication customer churn dataset (Telco 2), https://www.kaggle.com/competitions/customer-churn-prediction-2020

- Telecommunication customer churn dataset (Telco 3), https://www.kaggle.com/datasets/royjafari/customer-churn

- Insurance customer churn dataset, https://www.kaggle.com/datasets/k123vinod/insurance-churn-prediction-weekend-hackathon

The code is available at GitHub and Zenodo:

- https://github.com/mahayasa/ctgan-enn-cs

- I Nyoman Mahayasa Adiputra. (2025). mahayasa/ctgan-enn-cs: ctgan-enn-cs v1 (ctganenncs). Zenodo. https://doi.org/10.5281/zenodo.14892530

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
