# Peer review of "Optimized customer churn prediction using tabular generative adversarial network (GAN)-based hybrid sampling method and cost-sensitive learning"

_PeerJ Computer Science, doi:10.7717/peerj-cs.2949_

## Round 0.1 · original submission · Major Revisions

Please respond to the comments of the reviewers

·

Basic reporting

The manuscript is presented in English and generally adheres to professional standards of expression and courtesy. However, there are instances where clarity and smoothness of the text would benefit from rephrasing for easier reading and understanding. For example, the manuscript uses both "non-churn" and "majority class" as terms interchangeably.
Areas to improve: Designating one term for every concept and not switching from one term to the next would minimize confusion significantly. Additionally, there are some very long sentences that, for readability, could easily be divided into separate sentences. Besides, the motivation for the use of CostLearnGAN is not well explained. Authors should provide more information on the limitations of the existing methods, such as SMOTE and ENN, and how their framework will handle such issues. Furthermore, the research gap should be highlighted by the author. While the introduction does mention relevant prior works and outlines the technical challenges, it does not put sufficient emphasis on the particular research gap or the specific advantages of the proposed method over existing techniques.

Experimental design

From the experimental design more details can be provided regarding biases or any ethical considerations in processing data concerning the fairness and applicability of cost-sensitive learning to real-world customer datasets. The general setup is well-documented, but the rationale and protocol behind choosing specific class weight ranges and hyperparameters when tuning the model could be further elaborated to explain how these choices would affect performance.

Validity of the findings

The integration of CTGAN-ENN with cost-sensitive learning is indeed novel, and the results are impressive. The author should explicitly explain how this article improves GAN-based hybrid sampling compared to previous techniques, such as SMOTE or GBHS (Zhu et al., 2022). The limitations of the study be elaborated more clearly; for example, the real data point may get lost as a result of under-sampling while anomalies in the case of CTGAN result in class boundary distortions. Besides, future directions should be incorporated beyond modifying ENN, for instance, extending the framework in unsupervised settings or even adapting it for non-tabular data. Moreover, discussion on Type I/II errors for hypothesis testing and how they can impact the measurements of the results on robustness. Provide sensitivity analyses of CostLearnGAN performance under different data imbalance ratios. The discussion can be extended to cover broader open questions, such as whether CostLearnGAN will be able to scale up to real-time applications or adapt to other domains different from customer churn prediction. This will help give a wider perspective on where future research is headed.

Additional comments

These scatterplots and other performance comparisons are useful but lack interpretive captions for a non-expert reader. The authors need to go further in pointing out the implications of such results and, for example, in what ways CostLearnGAN might be ported to other domains. Such refinements will go a long way in allowing this manuscript to make its best contribution to the field.
Recommendation
1. Expand the discussion on motivation, research gap, research questions, and objectives in the introduction.
2. Provide more detailed descriptions of preprocessing and implementation steps for reproducibility.

·

Basic reporting

The authors state that they propose a new method that incorporates improvements both at the level of data preprocessing (GAN-based oversampling) and at the level of the classification algorithm (cost sensitive learning).

However, the text of the paper completely lacks a description of how cost-sensitive learning was implemented, how the cost of misclassification was determined, etc... Standard classification metrics (ROC AUC, F1, ...) are used, although in the case of cost-sensitive learning these metrics should consider the cost of error (see, e.g., https://doi.org/10.1016/j.eswa.2019.06.009).

GAN-based oversampling is not described, it is simply stated that it has been used. There are many questions regarding GAN-based oversampling, in particular the generation of both categorical and numerical features (see, e.g. https://doi.org/10.1016/j.eswa.2021.114582 ). These issues are not disclosed.

The introduction and literature review do not sufficiently describe the context of the problem under investigation (see though the references to the two papers cited above).

All this (as well as several smaller deficiencies, see below) leads to the conclusion that the article lacks scientific novelty and should be rejected.

Experimental design

No comment

Validity of the findings

No comment

Additional comments

1. Introduction (page 6, line 61). Unclear statement: ‘Imbalanced data occurs when one of the classes is significantly different from the other class’. It is fine when the classes are different. An imbalance occurs when the number of instances of different classes differs.

2. Introduction (p.6, l.64). The class overlapping is not solely a consequence of oversampling.

3. Only sufficiently weak classifiers (DT, LR, SVM) are considered and just single hyperparameter is optimized (class balance). Obviously, such models always leave room for improvements. For a complete picture it is necessary to investigate more productive algorithms (e.g. boosting)

4. P.9, L. 182: variables x and y are not defined in the text.

5. P. 10, L. 216: The data are insufficiently described (acquisition, preprocessing, features...).

Reviewer 3 ·

Basic reporting

1. The introduction should be extended to cover the background and the main subject of the study. A brief discussion on customer churn prediction, as well as domains of application, is needed to put the introduction in the context of the current research in the domain
2. Although the authors mentioned in the introduction section the advancement in research on churn prediction, they failed to precisely identify the gap in the existing works that their study intends to fill. Imbalanced and overlapped data are general problems in churn prediction that have been addressed in the existing work. Also, the motivation for this research is not clearly stated. Authors should clearly state the problem in the existing churn prediction models they are tackling and what motivates addressing the problem.
3. The authors need to highlight the specific contributions of this study to the customer churn prediction domain
4. Overall, the language used should be improved. There are some grammatical errors and ambiguous statements identified in the manuscript, such as:
a. In recent years there have been several research to overcome imbalanced an overlapped data problem in customer churn prediction.... (lines 68-69)
b. "Recent research works on cost-sensitive learning (CSL) have been done to overcome imbalanced problems in machine learning classification tasks" (lines 73-74). what do the authors mean by "imbalanced problems"? I suggest using the proper term, data imbalance problem(s), for clarity.
c. The classical algorithms were chosen because achieved faster time consumption than ensemble machine learning (lines 93-94)
d. Their experiment results improved the generalization performance of SVM (lines 144-145)
e. their method achieved more effective for a variety of misclassification costs (lines 148-149)
f. Classicaludies show that cost-sensitive learning in classical machine learning results in satisfying performance (158-159)
and so on...
5. the use of bullet to list steps under experimental setting can be replaced with roman numbers.

Experimental design

1. The description of the method used is insufficient. A detailed explanation of the CTGAN synthetic data generation as well as the description and function of the ENN component of the hybrid model should be provided.
2. The class weight hypeparameter optimization method is also not detailed enough. Is the optimization done using the cross-validation method or another search-based hyperparameter optimization method? What cost-sensitive performance metric(s) was used as the objective function for selecting optimal parameter settings for the majority and minority classes, and why? Did the authors consider trade-offs between performance and cost? These questions should be adequately answered to present a strong argument for the proposed method.
3. It is clear from the authors description of the proposed method that it addresses the data imbalance problem. However, discussion on how the proposed method addresses the data overlapping problem is also crucial (authors claimed to address the two problems) and should be presented under the material and method section.
4. Table 2 adds no value to the work, the parameter is the same for all algorithms; it should be removed. The information in the table can be presented in a single statement.

Validity of the findings

Comparisms with the state-of-the-art approaches should be extended to include more recent works. This should be presented clearly in a table.

Additional comments

The authors proposed a hybrid data-level and algorithm-level enhancement to the customer churn prediction model using a hybrid sampling method composed of conditional tabular GAN and Edited Nearest Neighbor for synthetic data generation and balancing, and a cost-sensitive learning approach for fine-tuning class weight hyperparameter. These enhancements yielded improved performance of the classical machine learning models and produced better results in different domains with adequate robustness. However, the description of the methods and reporting is highly lacking in terms of depth and clarity which author need to revise adequately to rasie the standard of the manuscript.

---

## Round 0.2 · Major Revisions

Dear Authors,

The manuscript has been entrusted to me for editorial oversight, and it is evident that one of the reviewers has expressed significant issues and criticisms with the revised version.It is strongly recommended that these concerns be addressed and that the article be resubmitted following the requisite revisions.

Best wishes,

·

Basic reporting

The authors have addressed some of the reviewers' comments, but the paper is still far from being ready for publication.

1. The text is unclear and often technically incorrect, many problems with English. Revision and proofreading of the text are recommended. For example, these are just a few of the problems in the first two pages:

• Not clear statement (line 70): ‘But oversampling is not the only reason that caused overlapped data. natural variability and noisy data.’
• I suggest adding a few references when SMOTE as a local method and cost-sensitive learning (CSL) are discussed (line 95 and below).
• The few first sentences in paragraph beginning on line 110 repeat the content of the preceding paragraph.
• Incorrect statement, line 146: ‘It works by selecting a minority class…’. Must be ‘It works by selecting an example of minority class.’

In fact, such inaccuracies are found throughout the whole text

2. I think the Introduction should be rewritten again to show the context, the limitations of existing approaches and the authors' contributions. Each critical assertion should be supported by references to scientific papers.

3. One of the main problems is that the authors insist that they are following the CSL framework, but in fact just class weighting is used (and not quite correctly, see below). This is similar to CSL, but a different method. CSL should calculate the cost of misclassification either for a class or for a particular instance.

4. Still no mention of the use of GAN. At least what loss function is used, how are discrete features generated?

Experimental design

5. The proposed framework is very poorly described

• I didn't understand the role of cross-validation in the CostLearnGAN framework (line 230). I think it's part of the experiment procedure.
• What are weights (line 238) for? Is it a substitute for the real cost value? Then this method does not belong to CSL.

6. Algorithm 1: I assume that a classifier should be trained at each weighting iteration. But this step is missing.

7. Hyperparameters tuning (line 304): The authors write that they use Optuna (which supports many Bayesian methods), but Algorithm 1 is something like two sequential grid search procedures. Using Optuna, you can tune the x and y weights simultaneously.

8. Another obvious question - why should you set weights for minor and major classes at the same time? It is enough to find only the weight of the minor class, while the weight of the major class is always equal to 1. Table 7 confirms this assumption since there are no x, y pairs in which both values are greater than 0 (and what does 0 mean in this context?)

Validity of the findings

9. What are SM, SE, WP, WE and CT (competing methods) in Tables 2-5? Abbreviations are given at the end of the article, but it is better to do it at the beginning, because without them the text will be incomprehensible.

---

## Round 0.3 · Minor Revisions

Dear Authors,

While there have been improvements to the manuscript, we strongly recommend that you address the concerns and criticisms of Reviewer 4 and resubmit your paper once you have updated it accordingly.

Best wishes,

Reviewer 4 ·

Basic reporting

The manuscript is well-organized and follows standard academic structure: abstract, introduction, literature review, methods, experiments, results, and conclusion.Relevant literature on GAN-based oversampling, cost-sensitive learning (CSL), and hybrid sampling methods is sufficiently cited. The motivation behind the CostLearnGAN framework is now more clearly explained, including gaps in previous GAN or CSL-only methods.
- The revised version significantly improves clarity compared to the original submission. However, the manuscript still requires further English proofreading, particularly in the introduction, methodology, and results sections. While most key ideas are now understandable, there are lingering grammar and syntactic issues that may hinder readability for non-native readers (e.g., “do not achieve a satisfactory performance” or “it has not performed well with classical machine learning algorithms”).
- Figures and tables are informative and clearly formatted. The inclusion of ANOVA and Tukey post-hoc tests (Table 6), confusion matrix (Table 8), and robustness metrics strengthens the credibility of results. Still, figure captions could be more descriptive, and color scales in decision boundary figures (Figs. 2–7) would benefit from clearer legends.

Experimental design

The paper presents a well-designed hybrid framework combining CTGAN-ENN (oversampling + noise removal) with cost-sensitive learning using class weight optimization via Optuna.Data preprocessing is clearly described. The authors use 6 publicly available datasets, 5 classifiers, and 6 baseline sampling methods, making the setup robust.The integration of GAN-generated tabular data with cost-sensitive hyperparameter tuning is a meaningful contribution. The decision to tune both class weights (x, y) for churn/non-churn classes simultaneously is explained and defensible, though the reviewer raised valid points about the simplicity of fixing one weight at 1.

Validity of the findings

The proposed method (CostLearnGAN) achieves best results in 19–20 out of 30 test scenarios for F1, AUC, and G-Mean. Robustness metrics (Section 6) and statistical validation using ANOVA + Tukey HSD provide strong empirical support. Class weight tuning is shown to significantly improve minority class recall, which is critical for churn detection.The authors now acknowledge limitations such as:

- Sensitivity to under-sampling that may remove relevant minority instances.

- Difficulty tuning hyperparameters for different data distributions.
- Consider clarifying your use of CSL terminology (e.g., distinguish between class weighting and instance-level cost).

---

## Round 0.4 · Minor Revisions

Dear Authors,

Two reviewers suggest minor revision. We encourage you to address the concerns and criticisms of the reviewers and resubmit your paper once you have updated it accordingly.

Best wishes,

·

Basic reporting

There remain comments that I made to earlier versions of the paper:

- The authors insist that they are using cost-sensitive learning, when in fact they are just weighting instances of the minor class.

- The procedure for determining optimal weights (Algorithm 1) is inefficient, it would be quite sufficient to determine the weight of the minor class only, Table 7 confirms this.

Nevertheless, if other reviewers and the editorial board consider these comments to be irrelevant, I am willing to agree.

Experimental design

N/A

Validity of the findings

N/A

Additional comments

N/A

Reviewer 4 ·

Basic reporting

This manuscript proposes a hybrid framework called CostLearnGAN, combining Conditional Tabular GAN (CTGAN) with Edited Nearest Neighbor (ENN) under-sampling and cost-sensitive learning via Optuna-driven hyperparameter tuning. The goal is to enhance classical machine learning performance in customer churn prediction under imbalanced and overlapped data scenarios. The revision represents a substantial improvement over earlier versions. The structure is sound, methodology is rigorous, and experimental results are well presented and statistically validated. The authors have addressed previous reviewer concerns thoroughly, especially in clarifying the motivation, expanding on the robustness evaluation, and improving the clarity of the figures.
While clarity has improved, some grammatical and syntactic issues remain, particularly in the introduction and methodology sections (e.g., awkward phrasing like “do not achieve a satisfactory performance”). Further proofreading by a native English speaker is advised.

Experimental design

The proposed method is tested on six public datasets with varying degrees of class imbalance. The authors have responded constructively to suggestions on simplifying class weight tuning and acknowledge this as a future improvement direction.

Validity of the findings

CostLearnGAN consistently outperforms baseline techniques across multiple metrics: F1-score, AUC, and G-Mean. Statistical tests confirm the significance of improvements. Limitations such as sensitivity to under-sampling and hyperparameter tuning are appropriately acknowledged.

---

## Round 0.5 · accepted · Accept

Dear Authors,

Thank you for addressing the reviewers' comments. Your manuscript now seems sufficiently improved and ready for publication.

Best wishes,

Reviewer 4 ·

Basic reporting

All revisions requested by the reviewers have been thoroughly addressed, and the manuscript has been updated accordingly. The current version reflects all necessary corrections and improvements. Therefore, the manuscript is now suitable for acceptance in its present form.

Experimental design

The proposed method has been tested on six public datasets with varying degrees of class
imbalance. The results are given with well-known metrics. This section is sufficient.

Validity of the findings

My review about this seciton is positive in previous review.